# Co-Disposal of Coal Gangue and Red Mud for Prevention of Acid Mine Drainage Generation from Self-Heating Gangue Dumps

**Zhou Ran** [1,2], **Yongtai Pan** [1,2] **and Wenli Liu** [1,*]

[1] School of Chemical and Environmental Engineering, China University of Mining and Technology, Beijing 100083, China; rzhou9112@163.com (Z.R.); panyongtai@cumtb.edu.cn (Y.P.)

[2] Engineering Research Center for Mine and Municipal Solid Waste Recycling, China University of Mining and Technology, Beijing 100083, China

[*] Correspondence: liuwenli08@163.com; Tel.: +86-10-6233-9883

**Abstract:** The seepage and diffusion of acid mine drainage (AMD) generated from self-heating coal gangue tailings caused acid pollution to the surrounding soil and groundwater. Red mud derived from the alumina smelting process has a high alkali content. To explore the feasibility of co-disposal of coal gangue and red mud for prevention of AMD, coal gangue and red mud were sampled from Yangquan (Shanxi Province, China), and dynamic leaching tests were carried out through the automatic temperature-controlled leaching system under the conditions of different temperatures, mass ratios, and storage methods. Our findings indicated that the heating temperature had a significant effect on the release characteristics of acidic pollutants derived from coal gangue, and that the fastest rate of acid production corresponding to temperature was 150 °C. The co-disposal dynamic leaching tests indicated that red mud not only significantly alleviated the release of AMD but also that it had a long-term effect on the treatment of acid pollution. The mass ratio and stacking method were selected to be 12:1 (coal gangue: red mud) and one layer was alternated (coal gangue covered with red mud), respectively, to ensure that the acid-base pollution indices of leachate reached the WHO drinking-water quality for long-term discharge. The results of this study provided a theoretical basis and data support for the industrial field application of solid waste co-treatment.

**Keywords:** coal gangue; red mud; prevention; acid mine drainage; dynamic leaching

## 1. Introduction

Coal gangue is a by-product of coal mining and washing and its discharge amount accounts for approximately 10–15% of total coal production [1,2]. The coal gangue dump is formed by the direct storage of coal gangue and occupies a great deal of farmland and arable land resources [1,3]. The active components contained in coal gangue, such as combustible carbon, pyrite, and heavy metal sulphide, may induce spontaneous combustion in the weathering process. The four stages of spontaneous combustion of coal gangue are: oxygen combining with coal gangue through physical and chemical adsorption; the combined oxygen and pyrite in coal gangue undergoing chemical and microbial catalytic (*Thiobacillus ferrooxidans*) oxidation and emitting heat; the combustible carbon being rapidly oxidized and emitting heat at an accumulation temperature of 80–90 °C; and combustion occurring when the temperature reaches the fire point of the combustible carbon. The critical factors for causing the spontaneous combustion of coal gangue are combustible substance, oxygen, and thermal storage environment. The spontaneous combustion of coal gangue usually occurs at a depth of 2 m from the surface of the coal gangue and causes the release of a large quantity of toxic and harmful

gases, like CO, $H_2S$, $SO_2$, and $NO_x$ into the atmosphere [4–7]. The soluble salt in coal gangue and the soluble secondary sulfate formed by natural weathering are dissolved and leached under the action of precipitation and leaching to generate acid mine drainage (AMD), which is harmful to the soil, river, and groundwater systems. In particular, a considerable amount of toxic metalloids (arsenic (As), selenium (Se), and antimony (Sb)) and a minute amount of heavy metals (chromium (Cr), copper (Cu), lead (Pb), and zinc (Zn)) are transformed from sulfide form to soluble sulfate form by oxidation and contaminate soils and groundwater [8–14].

Many studies have been conducted on the prevention and control of environmental pollution caused by separate storage of coal gangue. The main options for remediating AMD are divided into active and passive treatments [6,15–22]. The active remediation, that is, adding the neutralizer, adsorbents, and sulfate-reducing bacteria into the collected wastewater, both increases the pH value of the wastewater through acid-base neutralization and reduces the content of sulfates by the adsorption and precipitation of heavy metals [23–31]. Another treatment is to flow the AMD through the artificial wetland, which effectively and continuously improves the quality of AMD by adsorption, filtration, and precipitation [32,33]. In addition, the recycling of wastewater, resource recovery of metals and rare earth resources from AMD are obtained by conducting electrodialysis, membrane separation, and nanofiltration, respectively [34,35]. However, the active treatment of AMD derived from coal gangue needs to meet two requirements: small water flow, and tiny acidic fluctuation. A multi-stage drainage treatment process should be adopted when the AMD has a high concentration of dissolved heavy metals. Moreover, the treatment produces large volumes of sludge.

The principal aim of passive treatment is to prevent the AMD produced by the oxidation of pyrite in coal gangue, and it can be obtained by removing sulfide, isolating water and air, directly passivating pyrite, and adding antibacterial agents and alkaline components [36]. The risk of AMD leaching was decreased from the source by removing pyrite from fine-grained coal gangue during the bubble flotation process [37–39]. Meanwhile, the low-sulfur clean coal was separated from coal gangue by the oil gathering process [40,41]. By covering coal gangue with soil, desulfurized tailings, activated sludge, and oxygen-consuming organic matter, or establishing the gangue tailing pond underwater, the coal gangue was prevented from contact with air, which weakened the chemical oxidation [18,37,42–44]. The carrier-microencapsulation was proposed to passivate pyrite by forming a barrier on the mineral surface [45,46]. In addition, the chemical and microbial catalytic oxidation (*Thiobacillus ferrooxidans*) of pyrite in coal gangue was weakened by mixing with fly ash, quicklime, and limestone, or spraying antibacterial agents on coal gangue [47–50]. The prevention process requires neither continuous alkali addition nor sludge retreatment, thus, it shortens the time of the treatment cycle and provides more economic benefits compared with active treatment [51].

Red mud, as a type of red silt-like solid waste, is generated from alumina production, in which bauxite is the raw material. Typically, about 1 to 2 tons of red mud are generated from 1 t of alumina production, and the annual discharge of red mud worldwide is approximately 150 Mt [52,53]. The comprehensive utilization rate of red mud is less than 15%, and the remaining part of red mud is disposed of by deep-sea dumping or dam storage [54,55]. The particle size of red mud particles is very small, of which more than 90% of particles have a size range from 5 to 75 μm. Red mud generates a large amount of fugitive dust after being air-dried and disintegrated, which increases the content of inhalable particulate matter in the atmosphere [56]. Red mud is a highly alkaline (pH = 10–12.5) and intensely sodic (exchangeable sodium percentage, or ESP, from 70% to 90%) material [57]. Alkaline components in red mud include the adsorption alkali and bound alkali. Among them, the adsorption alkali (free alkali) exists in the forms of $Na_2O \cdot Al_2O_3$, $Na_2CO_3$, $NaHCO_3$, $NaAl(OH)_4$, $NaAlO_2$, $Na_2SO_4$ and $KOH(H_2O)_4$, which can be directly dissolved and leached, while the bound alkali is composed mainly of calcite ($CaCO_3$), sodalite ($Na_6Al_6Si_6O_{24} \cdot 2Na_2X$), hydrated garnet ($Ca_3Al_2(SiO_4)_x(OH)_{12-4x}$) and tricalcium aluminate ($Ca_3Al_2(OH)_{12}$), which is dissolved through a chemical reaction [58–61]. Leaching of bauxite residues not only results in alkaline pollution and excessive fluoride content in groundwater but causes salinization and consolidation of the soil [62].

Therefore, red mud added to prevent acid pollution caused by coal gangue could be a potential method for solid waste's co-treatment projects. The present study focused on the release characteristics of the acid contaminant generated from self-heating oxidation of coal gangue. Meanwhile, during the co-disposal of red mud and coal gangue, the leaching regularities of acid pollutants were researched through the automatic temperature-controlled leaching system, which was designed to simulate the conditions of rapid oxidation precipitation, and leaching. Moreover, an optimal co-disposal program of red mud and coal gangue was proposed to ensure that water meets the quality standards so that it could be discharged outward.

## 2. Materials and Methods

### 2.1. Samples Preparation

The coal gangue and red mud samples were collected from the No.1 Gangue Dump of Yangquan Coal Industry and the red mud Reservoir of Zhaofeng Alumina Plant, respectively, both in Shanxi Province, China. The fresh coal gangue samples were the discharge of raw coal after washing, and their original particle size composition was >25 mm 61.26 wt%, 13–25 mm 23.51 wt%, and <13 mm 15.23 wt%, respectively. The pyrite in coal gangue presented in the single crystal, disseminated, and stripped states. When used directly without any pretreatment, the original sample would have an immense effect on the comparative analysis of results for the huge fluctuation of the sample composition. Therefore, the original coal gangue samples were ground into leaching samples, whose diameter was 1 mm, while the red mud samples obtained from the settled and thickened alumina tailings slurry through a quick-opening filter press contained 39.84 wt% water. To facilitate the co-disposal test, the red mud samples derived from the filter process were air-dried, crushed, and sieved into agglomerated samples with a diameter of 1 mm, which had the same physical and chemical properties as the original red mud samples [63].

### 2.2. Mineralogical Determination

The proximate analysis (M, Ash, VM, and FC) of the coal gangue samples (1 mm) was conducted according to ASTM D 3173-03 (Moisture, M), ASTM D 3174-04 (Ash), ASTM D 3175-07 (Volatile matter, VM), and the Fixed carbon (FC) was determined by subtracting the sum of M, Ash, and VM from 100%. The ultimate analysis was performed on an elemental analyzer (Vario MACRO CHNS, Elementar) with an oxygen kit. The mineralogical compositions of the samples were studied using a D/max-2550 X-ray diffractometer (Rigaku Corporation, Tokyo, Japan) with Cu-K$_{\alpha 1}$ radiation operating at 40 kV and 40 mA. The samples were scanned over 3–90° 2-theta with a step size of 0.02° 2-theta and a counting time of 1 s. The mineral content of the samples semi-quantitatively analyzed by the reference intensity ratio (RIR) method was obtained with the help of MDI Jade6.5 software. An XRF-1800 X-ray fluorescence spectrometer (Shimadzu Corporation, Kyoto, Japan) with Rh radiation and 60 kV(Max) X-ray tube pressure was used to measure the chemical composition of the samples. The extraction characteristics of the samples were analyzed according to ASTM D3987-12. The extractant was the reagent water, of which the volume equalled in milliliters to 20 times the mass in grams of the samples. Besides, the particle sizes of red mud samples (1 mm) were determined by laser particle analyzer (LS-POP(VI), Zhuhai, China). All of the mineralogical indicators were measured 3 times, and the data was analyzed for standard deviation (SD).

### 2.3. Leaching Experimental Device

To truly simulate the physical and chemical conditions of the self-heating coal gangue dumps' precipitation, an automatic temperature-controlled leaching system was designed for this research, as is shown in Figure 1.

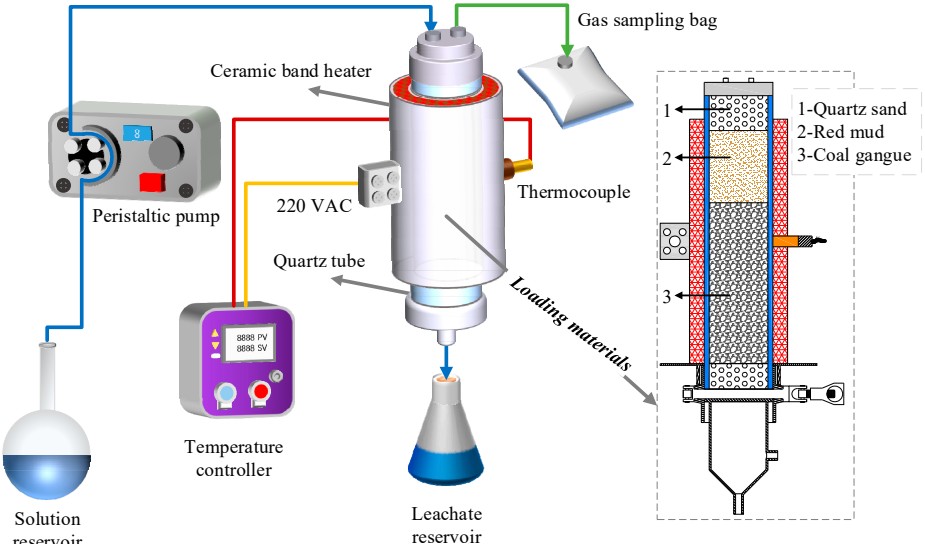

**Figure 1.** Schematic diagram of the automatic temperature-controlled leaching system and material loading.

The system was mainly composed of the quartz tube, ceramic band heater, temperature controller, peristaltic pump, gas sampling bag, solution reservoir, and leachate reservoir. The size of the quartz tube was $\Phi$ 70 mm $\times$ 5 mm, $h$ = 300 mm, and the quartz tube and its inside samples were heated by the ceramic band heater tightly clamped outside. The temperature controller with a control range of 0–300 °C was used to detect and control the working temperature of the ceramic band heater. The spray liquid in the solution reservoir was sent to the quartz tube by the peristaltic pump. The gas sampling bag collected toxic and harmful gases generated from the heating process, and the leaching solution was collected by the leachate reservoir.

## 2.4. Dynamic Heating and Leaching

To explore the prevention effect of red mud on the acid pollution derived from coal gangue dumps in a self-heating state, it was necessary to grasp the release characteristics of the coal gangue acid contaminant at different temperatures. Under the condition of the fastest release rate of acidic components in coal gangue, dynamic leaching tests were performed on the co-disposal of red mud and coal gangue to research the co-disposal effect.

Firstly, the releasing characteristics of acidic pollutants derived from coal gangue were studied. The coal gangue samples (1 mm) were separately packed into the quartz tube according to the loading method shown in Figure 1. The samples were filled into the quartz tube by gravity accumulation without external pressure. The mass of the coal gangue sample in the middle layer was set as 600 g, and the height of the coal gangue layer was 60 mm. The upper and lower layers were paved with 10–20 mesh quartz sand to ensure the spray solution's uniform infiltration and filter out the coarse particles in the leaching solution. The spray intensity of the peristaltic pump was set to be 6 mL·min$^{-1}$ according to the rain capacity in Yangquan area (Shanxi Province, China), and the spray liquid was deionized water. At $t$ = 0 h, the materials in the quartz tube were initially wetted, and the wetting process was stopped when a 250 mL leaching solution had been collected. Then, temperatures of the ceramic band heater were set at 50 °C, 100 °C, 150 °C, and 200 °C to heat the coal gangue sample. The quartz tube was heated at intervals of 10 h for spraying operation until 250 mL leaching solution had been collected, and the cumulative heating time of single factor test (e.g., 50 °C) was 160 h. The pH value, electrical conductivity (EC) value, sulfate (SO$_4^{2-}$) concentration, oxidation-reduction potential (ORP), total dissolved solids content (TDS), and acid neutralization potential (ANP) of the leachate were detected immediately. These indicators of the leachate were repeatedly tested three times, and the

average value was taken to reduce the detection error. The apparatus and methods are shown in Table 1, and the dynamic heating and leaching process is shown in Figure 2.

**Table 1.** The apparatus and methods for detecting pollution indices of leachate.

| Parameter | Apparatus and Methods |
|---|---|
| pH | FE28-standard pH meter (Mettler Toledo, Küsnacht, Switzerland) |
| EC | DDS-11A electrical conductivity meter (Lei Ci, Shanghai, China) |
| $SO_4^{2-}$ | Water quality-Determination of sulfate-Gravimetric method (GB11899-89) |
| ORP | Orion 3 Star mV Detector (Thermo Scientific, Waltham, MA, USA) |
| TDS | TDS-5 m (Greensky, Hangzhou, China) |
| ANP | Acid-base neutralization titration |
| Mineral composition | D/max-2550 X-ray diffractometer (Rigaku Corporation, Tokyo, Japan) |

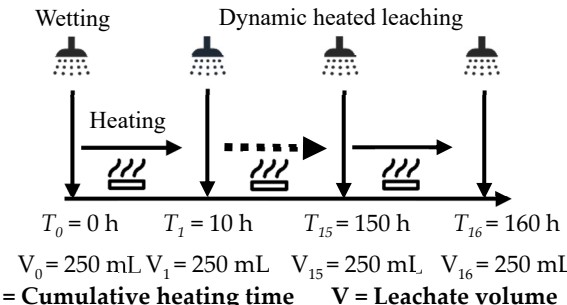

**Figure 2.** Schematic diagram of dynamic heating and leaching process.

Additionally, at the temperature corresponding to the fastest release rate of acidic components of coal gangue, the dynamic leaching tests of the co-disposal of the coal gangue and red mud were carried out under the conditions of different mass ratios (12:1, 8:1, 5:1, and 3:1) and storage methods (alternating 1 layer, 2 layers, 5 layers, and uniform mixing). In the mass ratios study, 600 g coal gangue (thickness: 60 mm) was used as the mass benchmark and covered with the red mud of different mass, and the filled red mud masses were 50 g (40 mm), 75 g (60 mm), 120 g (96 mm), and 200 g (160 mm), respectively, corresponding to the mass ratios of 12:1, 8:1, 5:1, and 3:1. Besides, in the storage method study, the mass ratio of coal gangue to red mud remained unchanged (e.g., 12:1 or 600 g:50 g), and the certain mass of coal gangue and red mud was equally divided into 1, 2 and 5 parts, respectively, corresponding to the storage method of alternating 1 layer, 2 layers, and 5 layers. Each part was filled into the quartz tube alternately in the order of coal gangue in the lower layer and red mud in the upper layer. As for the storage method of uniform mixing, the coal gangue and red mud samples were uniformly mixed by the mixer prototype (MR2L, France). The samples were filled into the quartz tube by gravity accumulation without external pressure, and the co-disposal dynamic heating and leaching mechanism was the same as the research method of the release characteristics of the acidic pollutant derived from coal gangue.

## 3. Results and Discussion

### 3.1. Properties of Materials

The proximate and ultimate analysis of coal gangue samples are shown in Table 2.

**Table 2.** Proximate and ultimate analysis of coal gangue (wt%, mean ± SD, n = 3).

| Proximate Analysis | | Ultimate Analysis | |
|---|---|---|---|
| Moisture, $M_{ad}$ [1] | 1.07 ± 0.02 | Carbon, $C_{ad}$ | 17.60 ± 0.06 |
| Ash, $A_{ad}$ | 72.70 ± 0.5 | Hydrogen, $H_{ad}$ | 1.36 ± 0.05 |
| Volatile matter, $VM_{ad}$ | 9.25 ± 0.02 | Nitrogen, $N_{ad}$ | 0.41 ± 0.03 |
| Fixed carbon, $FC_{ad}$ | 16.98 ± 0.41 | Sulfur, $S_{ad}$ | 4.50 ± 0.16 |

[1] ad: Air-dried statement.

The coal gangue had great potential of self-heating and spontaneous combustion, and the content of the $FC_{ad}$ and $C_{ad}$ accounted for 16.98 ± 0.41 wt% and 17.60 ± 0.06 wt%, respectively [64]. The potential acid production of coal gangue was 137.64 kg $CaCO_3 \cdot t^{-1}$ calculated by the standard Sobek acid-base counting test based on the 4.50 ± 0.16 wt% $S_{ad}$ [65]. The coal gangue sample was medium-sulfuric and had a strong potential for acid production.

XRD pattern of coal gangue revealed that the main minerals consisted of 31.88 ± 0.14% kaolinite, 26.78 ± 0.27% illite, 22.96 ± 0.20% quartz, 9.80 ± 0.37% pyrite and, 8.58 ± 0.09% calcite (Figure 3a). The content of the clay minerals (kaolinite, illite, and montmorillonite) exceeded 50%. This indicates that the sample is clay salt coal gangue and had a certain degree of acid neutralization potential, which could neutralize acidic components generated by coal gangue itself. As is shown in Figure 3b, the mineral composition of the red mud sample included 51.74 ± 0.35% hydrated andradite, 21.71 ± 0.21% calcite, 21.46 ± 0.05% hydrated sodalite, and 5.09 ± 0.05% hematite. The hydrated sodalites is the main alkaline component that accounts for the great potential of alkali production in red mud [66].

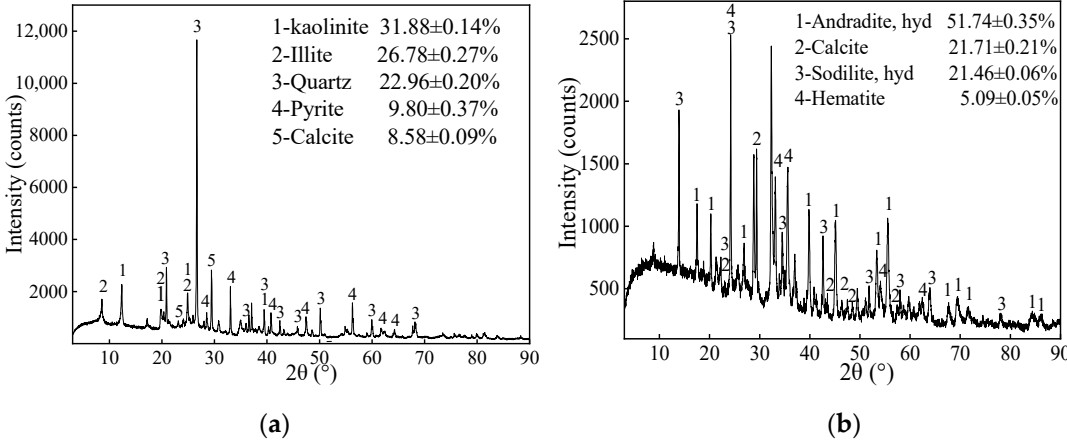

**Figure 3.** The XRD patterns of (**a**) coal gangue, (**b**) red mud, and the mineralogical compositions (%, mean ± SD, n = 3).

According to Table 3, the chemical composition of coal gangue was mainly $SiO_2$ and $Al_2O_3$ with contents of 49.81 ± 0.06% and 24.82 ± 0.13%, respectively. It showed that the main compound in the coal gangue sample was aluminosilicate. The total content of $Al_2O_3$ and $Fe_2O_3$ in red mud exceeded the total mass of $SiO_2$ and CaO, and the content of $Na_2O$ was 12.05 ± 0.07%. This indicated that the red mud sample was the bauxite residue of the Bayer process and was strongly alkaline.

**Table 3.** Chemical compositions of coal gangue and red mud (wt%, mean ± SD, n = 3).

| Material | Oxides | | | | | | | | | | | Loss |
|---|---|---|---|---|---|---|---|---|---|---|---|---|
| | $SiO_2$ | $Al_2O_3$ | $Fe_2O_3$ | CaO | $SO_3$ | $Na_2O$ | $Cr_2O_3$ | NiO | CuO | ZnO | PbO | |
| Coal gangue | 49.81 ± 0.06 | 24.82 ± 0.13 | 9.82 ± 0.05 | 6.39 ± 0.04 | 5.52 ± 0.03 | 0.45 ± 0.01 | 0.03 ± 0.002 | 0.01 ± 0.00 | 0.01 ± 0.00 | 0.02 ± 0.00 | 0.01 ± 0.00 | 27.2 ± 0.12 |
| Red mud | 20.43 ± 0.16 | 25.92 ± 0.09 | 14.62 ± 0.18 | 17.22 ± 0.08 | 2.61 ± 0.02 | 12.05 ± 0.07 | 0.06 ± 0.003 | 0.08 ± 0.002 | 0.009 ± 0.00 | 0.01 ± 0.00 | 0.01 ± 0.00 | 7.10 ± 0.16 |

The shake extraction indices of samples are shown in Table 4. The pH value and ANP of coal gangue were 7.25 ± 0.04 and −0.071 ± 0.003 g (CaCO$_3$)·L$^{-1}$, respectively, and it indicated that the sample presented as weakly alkaline during the shake extraction process. The coal gangue that has not been weathered and oxidized does not cause acid pollution to the environment. However, the red mud presented as strongly alkaline, in which the alkaline components were leached in a large amount during the extraction process, and the pH value was 11.11 ± 0.05. The EC values of coal gangue and red mud were 0.715 ± 0.009 and 0. 852 ± 0.006 mS·cm$^{-1}$, respectively, and the TDS were 581 ± 5 and 509 ± 4 mV. Since both the EC value and TDS reflected the concentration of soluble ions in the solution (natural water EC value, 0.102–2.079 mS·cm$^{-1}$), the result showed that both extracted solutions of the samples had little effect on the hardness of water [67]. Moreover, the extracted solutions of coal gangue and red mud presented as weakly oxidative for the ORP of 141.3 ± 4.2 and 8.52 ± 0.6 mV, respectively. Besides, the average particle diameter ($D_{(4, 3)}$) of the red mud sample was 5.74 ± 0.02 μm and the $D_{90}$ was 8.59 ± 0.02 μm, indicating that the sample had an extremely fine particle size (Figure 4). The fine particles cause the red mud to have uniformity of physical and chemical properties and low permeability.

**Table 4.** The shake extraction indices of coal gangue and red mud (mean ± SD, n = 3).

| Parameter | Unit | Material | |
|---|---|---|---|
| | | **Coal Gangue** | **Red Mud** |
| pH | | 7.25 ± 0.04 | 11.11 ± 0.05 |
| EC | mS·cm$^{-1}$ | 0.715 ± 0.009 | 0.852 ± 0.006 |
| ORP | mV | 141.3 ± 4.2 | 8.5 ± 0.6 |
| TDS | ppm | 581 ± 5 | 509 ± 4 |
| ANP | g(CaCO$_3$)·L$^{-1}$ | −0.071 ± 0.003 * | −0.52 ± 0.02 |

\* A positive value indicates that the acidic solution consumes alkali, while a negative value indicates that the solution is alkaline.

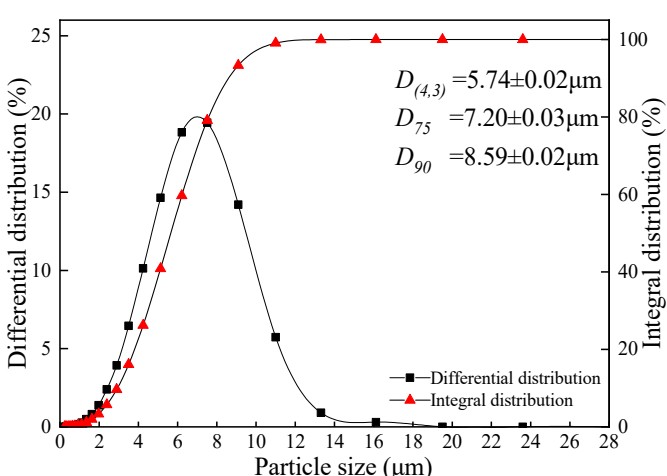

**Figure 4.** The particle size (mean ± SD, n = 3) distribution of red mud.

### 3.2. Release Characteristics of Acidic Contamination

The pH value and ANP of leaching solution derived from coal gangue corresponding to different dynamic heating and leaching tests varied as the cumulative heating time increased, as shown in Figure 5.

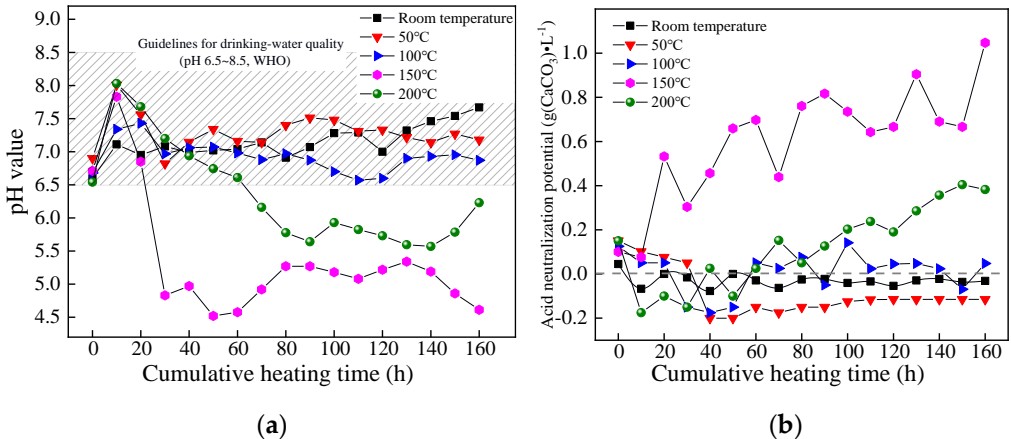

**Figure 5.** Variations of (**a**) pH value and (**b**) ANP of coal gangue leaching solution at different temperatures.

As is shown in Figure 5a, the variations of the pH value of coal gangue's leachate under different temperature conditions were similar as the cumulative heating time went by, that was, the pH value increased in the beginning, then decreased and eventually stabilized. When the pH value presented as stable ($t > 80$ h), the steady-state pH value decreased at first and then increased as the temperature increased, during which the minimum pH value was about 4.61 at 150 °C ($T_c$). The minimum pH value appeared to be strongly acidic and was out of the limit of the WHO guidelines for drinking-water quality (pH 6.5–8.5). This means that the acidic contamination of coal gangue was fully released under this temperature condition.

The main acid producing component in coal gangue is the pyrite; the related Equations (1)–(6) chemical reaction equations during the heating and filtration processes can be seen as follows.

The pyrite produces acids directly by oxidation under the condition of air and water [11]:

$$2FeS_2(s) + 7O_2(aq) + 2H_2O \rightarrow 2Fe^{2+}(aq) + 4SO_4^{2-}(aq) + 4H^+(aq) \tag{1}$$

$$FeS_2(s) + 14Fe^{3+}(aq) + 8H_2O \rightarrow 15Fe^{2+}(aq) + 2SO_4^{2-}(aq) + 16H^+(aq) \tag{2}$$

$$4Fe^{2+}(aq) + O_2(aq) + 4H^+(aq) \rightarrow 4Fe^{3+}(aq) + 2H_2O \tag{3}$$

The pyrite is oxidized by air without water:

$$FeS_2(s) + 11O_2(g) \rightarrow 2Fe_2O_3(s) + 8SO_2(g) \tag{4}$$

$$2SO_2(g) + O_2(g) \rightarrow 2SO_3(g) \tag{5}$$

$$4FeS_2(s) + 3O_2(g) \rightarrow 2Fe_2O_3(s) + 8S(s) \tag{6}$$

The rate of pyrite's oxidizing and production acid accelerates with the increase in external temperature under the condition of water [68,69]. However, when the temperature reaches a higher state ($T > 200$ °C), a large amount of water inside the coal gangue is evaporated, and it curbs the direct oxidization and acid production of pyrite. The main products of the pyrite oxidized by the air are sulfur oxides ($SO_x$) and sulfur (S), which means there is a decrease in the acid-soluble components [70–72].

In addition, the calcite and silicate minerals contained in the coal gangue sample can neutralize AMD derived from the coal gangue itself, and this causes the pH value to gradually increase in the early stage of dynamic leaching tests. The related Equations (7)–(9) chemical reaction equations during the AMD neutralization processes present as follows.

$$CaCO_3(s) + 2H^+(aq) \Leftrightarrow Ca^{2+}(aq) + CO_2(g) + H_2O \tag{7}$$

$$NaAlSi_3O_8(s) + 4H^+(aq) + 4H_2O \rightarrow Na^+(aq) + Al^{3+}(aq) + 3H_4SiO_4(aq) \tag{8}$$

$$KAlSi_3O_8(s) + 4H^+(aq) + 4H_2O \rightarrow K^+(aq) + Al^{3+}(aq) + 3H_4SiO_4(aq) \tag{9}$$

Figure 5b shows that the leaching solution's ANP was small under a low-temperature condition (room temperature–100 °C), which fluctuated in the range of −0.2–0.18 g $(CaCO_3) \cdot L^{-1}$ and had little potential to cause acid pollution, under a high-temperature condition (from 150 °C to 200 °C), however, the ANP gradually increased as the cumulative heating time increased. In the temperature range of this study (room temperature–200 °C), the ANP reached its highest level at 150 °C·($T_c$), indicating that the pyrite in the coal gangue produced the largest quantities of acidic components at $T_c$, which accords with the pH analysis results in Figure 5a. Meanwhile, it also indicated that the coal gangue sample continuously caused acid pollution for a long time under a suitable temperature condition.

The mineral composition of coal gangue samples had been analyzed after being heated separately. The X-ray diffraction pattern is shown in Figure 6.

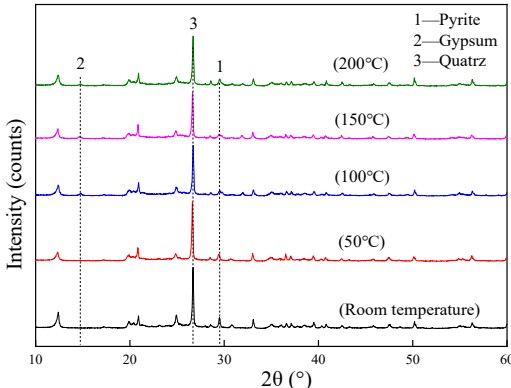

**Figure 6.** XRD patterns of the remaining coal gangue at different temperatures.

As the temperature increased, the intensity of the diffraction peaks of the pyrite in the remaining coal gangue weakened first and then strengthened, while that of the gypsum first strengthened and then weakened, and the change in quartz was not significant. The content of the pyrite was negatively correlated with the gypsum. The reason is that the sulfide in the coal gangue is heated and oxidized to produce sulfate, whereas part of the gypsum component in the remaining coal gangue is produced by the oxidation of pyrite. The pyrite in coal gangue has strong chemical activity. However, the generated gypsum has good chemical stability, and its coverage on the pyrite can prevent further oxidation and acid production.

### 3.3. Co-Disposal Prevention

Based on the research results of the acid pollution release characteristics of coal gangue in Section 3.2, the temperature of 150 °C ($T_c$) was chosen to carry out the dynamic leaching test through the co-disposal of red mud and coal gangue to prevent the acid pollution caused by the coal gangue. The tests were conducted under the condition of different mass ratios and storage methods to find a suitable storage solution.

### 3.3.1. Mass Ratio

pH Value

The coal gangue covered with the red mud was stacked in layers with different mass ratios (12:1~3:1) and then dynamically heated and leached. The pH value of the leaching solution changed as the cumulative heating time increased, as shown in Figure 7.

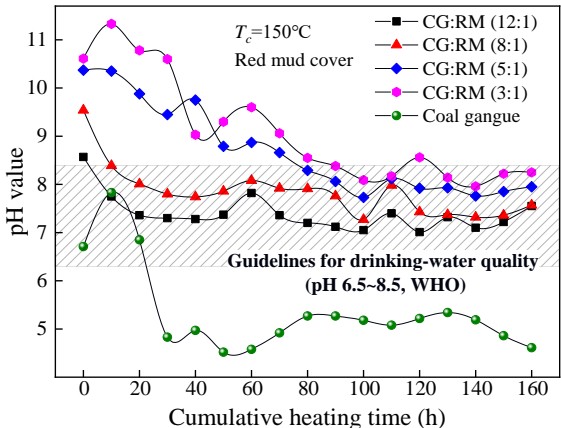

**Figure 7.** Variations of pH value of co-disposal leaching solution with different mass ratios.

　　The pH values of the leaching solution corresponding to different mass ratios (12:1~3:1) changed similarly as the cumulative heating time increased, that is, they gradually decreased and then remained stable. In the late stage of leaching (80–160 h), the steady-state pH value of the leaching solution was within the range of the drinking-water quality. This indicates that the addition of red mud can effectively alleviate the AMD generated from the coal gangue. The AMD is neutralized by the alkaline component in red mud as described by Equations (10)–(17) [66].

　　AMD is neutralized by the adsorption alkali:

$$NaHCO_3(s) + H^+(aq) \rightarrow Na^+(aq) + CO_2(g) + H_2O \tag{10}$$

$$Na_2CO_3(s) + 2H^+(aq) \rightarrow 2Na^+(aq) + CO_2(g) + H_2O \tag{11}$$

$$NaAl(OH)_4(s) + H^+(aq) \rightarrow Na^+(aq) + Al(OH)_3(s) + H_2O \tag{12}$$

$$KOH(H_2O)_4(s) + H^+(aq) \rightarrow K^+(aq) + 5H_2O \tag{13}$$

　　AMD is neutralized by the bound alkali:

$$CaCO_3(s) + 2H^+(aq) \rightarrow Ca^{2+}(aq) + CO_2(g) + H_2O \tag{14}$$

$$[Na_6Al_6Si_6O_{24}]\cdot[2NaX, Na_2X](s) \rightarrow 8Na^+(aq) + 6H_4SiO_4(aq) + 6Al(OH)_3(s) + 8X(OH^-, HCO_3^{2-})(aq) \tag{15}$$

$$2H_4SiO_4(aq) + 4H^+ \rightarrow Si^{4+}(aq) + H_2SiO_3(aq) + 5H_2O \tag{16}$$

$$Al(OH)_3(s) + 3H^+(aq) \rightarrow Al^{3+}(aq) + 3H_2O \tag{17}$$

　　At $t = 0$ h, the initial wetting solutions corresponding to each mass ratio were alkaline (pH > 8.5), and the initial pH value increased as the mass ratio decreased. This indicates that the more bauxite tailings are added, the stronger alkaline of the initial wetting solution would be. As shown in Equations (10)–(13), the adsorption alkali component in the red mud is particularly soluble and can be dissolved and leached in a large amount under the action of wetting spray, of which the leaching amount increases with the addition of red mud. How to effectively control the rapid dissolution of adsorption alkali in the early stage of leaching is a problem to be solved urgently.

　　During the entire process, the lower the mass ratio was (12:1→3:1), the higher the pH value of the corresponding leaching solution was. The pH value corresponding to the low mass ratio (5:1~3:1) was higher than that of the index limit of the drinking-water quality (pH 6.5–8.5) in the early stage of leaching (0–80 h). By contrast, the pH value for the high mass ratio (12:1~8:1) was within the range of the drinking-water quality during the entire process. The cumulative heating time required for the pH value to reach a stable state increased as the mass ratio decreased, in which the pH value corresponding to the mass ratio of 3:1 was 100 h. The more red mud was added into the stack, the longer the leaching

time that was required for the complete leaching of the adsorption alkali component. In the late stage of leaching (80–160 h), the steady-state pH value increased slowly with the decrease in the mass ratio for the leaching component of red mud due mainly to the bound alkali. Acid-base neutralizing between the acidic component produced by coal gangue and the bound alkali component derived from red mud takes place mainly by the displacement reaction, catalytic reaction, and hydration reaction as shown in Equations (14)–(17), of which the degree of the acid-base neutralization depends on the content of the acidic components produced by coal gangue.

ANP

The ANP of the leaching solution obtained from the co-disposal of red mud and coal gangue according to different mass ratios changed as the cumulative heating time increased, as shown in Figure 8.

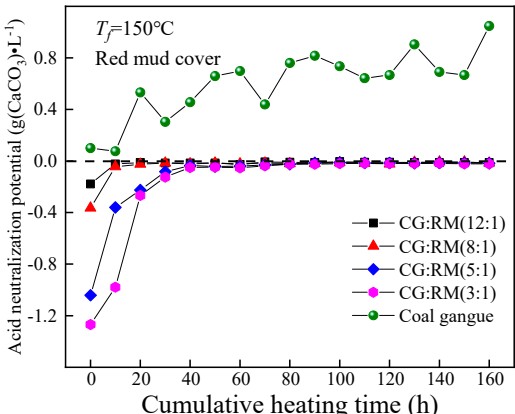

**Figure 8.** Variations of ANP of co-disposal leaching solution with different mass ratios.

Throughout the entire heating and leaching process, the ANP values corresponding to different mass ratios were all negative, and the leaching solution presented as alkaline. Additionally, the ANP increased rapidly and eventually approached 0 g (CaCO$_3$)·L$^{-1}$. This further demonstrates that the red mud could prevent the acid pollution caused by the self-heating oxidation of the separately-stored coal gangue. At $t = 0$ h, the ANP of the initial wetting solution increased with the increase in mass ratio (3:1→12:1). The ANP of the coal gangue wetting solution was 0.10 g(CaCO$_3$)·L$^{-1}$, while the mass ratio of 3:1 corresponded to the ANP of −1.27 g(CaCO$_3$)·L$^{-1}$. This indicates that the addition of red mud has a significant impact on the ANP of the leaching solution. The possible reason is that the rapid dissolution of adsorption alkali (free alkali) components in the red mud layer results in the strong alkalinity in the leaching solution.

In the late stage of leaching (40–160 h), the mass ratio had little effect on the ANP of the leaching solution, as the ANP of different mass ratios steadily converged towards 0 g(CaCO$_3$)·L$^{-1}$. Meanwhile, adding a small amount of the red mud was enough to decrease the ANP of the coal gangue leaching solution, for it was the bound alkali that neutralized the acidic product produced by coal gangue and the releasing amount of the bound alkali depended on the quantity of the acid product from coal gangue. Compared with the pH index of the leaching solution, the cumulative heating time required for the ANP to reach the steady-state is less than that of the pH value. This indicates that the complex composition of alkaline minerals in the red mud and its strong anti-acidity capacity could delay the decrease in the leaching solution's pH value [73].

EC Value

The EC value of the leaching solution obtained by the co-disposal of red mud and coal gangue in different mass ratios varied with the cumulative heating time, as shown in Figure 9.

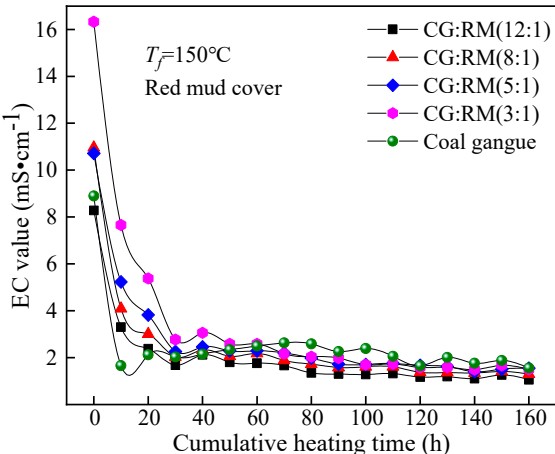

**Figure 9.** Variations of EC value of co-disposal leaching solution with different mass ratios.

The EC values of the leaching solutions corresponding to different mass ratios had a similar trend in the entire heating and leaching process, that was, they first decreased sharply, and then stabilized at about 2 mS·cm$^{-1}$ for a long time, which was higher than the range of general natural water's EC value (0.102–2.079 mS·cm$^{-1}$) [74]. This indicated that the co-disposal of red mud and coal gangue would increase the hardness of the surface water systems and the groundwater bodies after being leached. The EC value of the solution depends on the concentration of soluble ions and their salt content. The soluble mineral components contained in the coal gangue and red mud samples were relatively high and quickly dissolved and released in the early stage of leaching (0–30 h), while they were significantly reduced in the late stage of leaching (30–160 h). At $t$ = 0 h, the EC values of the initial wetting solutions corresponding to different mass ratios all reached the maximum, and the maximum value increased as the mass ratio decreased (12:1→3:1). When the mass ratio decreased from 8:1 to 3:1, the wetting solution was regarded as highly mineralized water (EC > 10 mS·cm$^{-1}$). The coal gangue contains a large number of clay minerals (kaolinite, montmorillonite, and illite), meanwhile, the red mud contains various forms of adsorption alkali. The clay minerals and the adsorption alkali components are quickly dissolved and leached under the action of initial wetting and spraying. The co-disposal of red mud and coal gangue can significantly increase the water hardness.

In the late stage of dynamic leaching (30–160 h), the EC values corresponding to the different mass ratios reached a stable state. The mass ratio had no significant effect on the steady-state EC value, for the concentration of soluble ions in the leaching solution depended on the acidic components produced by the coal gangue and the bound alkali component derived from the red mud. The total leaching amount of bound alkali depended on the content of acidic components produced by the coal gangue. Therefore, the mass ratio of coal gangue to red mud does not have a significant impact on the hardening water quality caused by long-term storage.

Therefore, the mass ratios of the co-disposal of coal gangue and red mud should be 12:1, for the pH of the leaching solution was within the range of drinking-water quality for a long time and the leaching solution had the least impact on the hardness of groundwater.

### 3.3.2. Storage Method

#### pH Value

The mass ratio of coal gangue to red mud was designed as 12:1, and the dynamic leaching tests were carried out under the condition of 150 °C. The pH value of leaching solution derived from the co-disposal of coal gangue and red mud based on the different storage methods (alternating 1, 2, 5 layers, and uniform mixing) varied with the accumulated heating time, as shown in Figure 10.

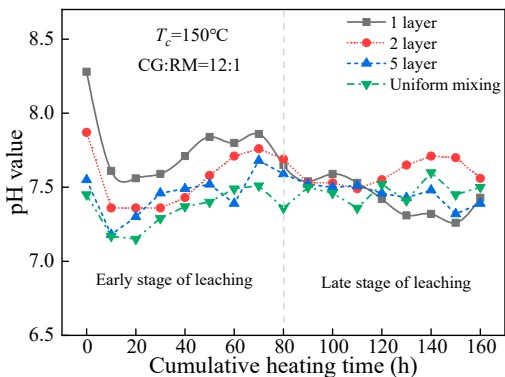

**Figure 10.** Variations of pH value of co-disposal leaching solution with different storage methods.

As shown in Figure 10, the storage method of coal gangue and red mud had no significant impact on the co-treatment of acid-base pollution throughout the entire heating and leaching process. The pH values of the leaching solution corresponding to the different alternate layers underwent similar changes, that was, they first decreased, then increased, and finally stabilized. The pH values did not exceed the drinking-water quality (pH 6.5–8.5) during the entire process, and the steady-state pH value approached 7.5 in the late stage of leaching (80–160 h).

At $t = 0$ h, the pH values of the initial wetting solutions reached the maximum, and the maximum values of pH decreased with the increase in the number of the alternate accumulation layers. It was the increasing number of the alternate accumulation layer that promoted the coal gangue being in contact with the red mud to strengthen the acid-base neutralization.

In the early stage of leaching (10–80 h), the pH value of the leaching solution corresponding to the various storage methods gradually increased. This indicated that the total alkaline concentration of red mud was higher compared with the acid concentration of coal gangue, for the leaching of the alkali in the red mud sample contained the adsorption alkali and the bound alkali at this stage. The bound alkali has a strong buffering capacity, which can fully react with the acidic components generated from coal gangue by displacement, catalytic, and hydration reactions. In the late stage of leaching (90–160 h), the pH value tended to be stable, and the steady-state pH value was not significantly affected by the number of alternate accumulation layers, for the residual content of adsorption alkali in the red mud sample was extremely small. Besides, the leaching alkali solution mainly contained the bound alkali in the late stage, and the amount of bound alkali leaching depended on the total amount of acid produced by the oxidation of coal gangue.

ANP

The ANP of the leaching solution varied with the cumulative heating time, as shown in Figure 11.

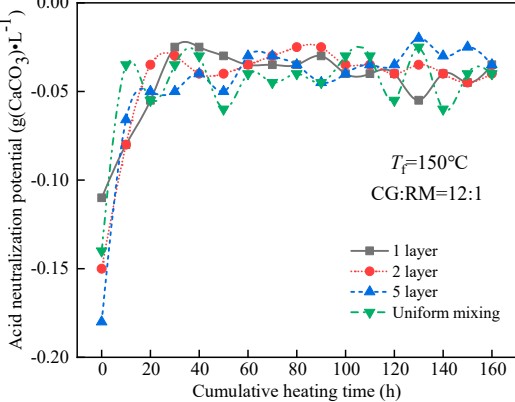

**Figure 11.** Variations of ANP of co-disposal leaching solution with different storage methods.

Figure 11 shows that the ANPs of the leaching solution corresponding to various stacking methods had a similar change during the entire heating and leaching process. The ANP reached a stable state after a rapid increase and the steady-state value approached −0.04 g (CaCO₃)·L⁻¹, which did not cause acid-base pollution to the surrounding water bodies. At $t = 0$ h, the ANP of the initial wetting solution was the smallest while alkalinity was the strongest. Meanwhile, the ANP decreased with the increase in the alternating stacking layers but the uniform mixing one increased exceptionally, for the ANP was determined by the acidic components of coal gangue, the alkaline components of coal gangue, and the adsorption alkali of red mud at the early stage of leaching. The alkaline components of coal gangue are mainly clay minerals, which leached in large amounts during the initial wetting, and the amount of leaching increases with the increase in the number of alternate accumulation layers of coal gangue and red mud.

During the late stage of leaching (40–160 h), the ANP of the leaching solution tended to be stable and was not significantly affected by the storage methods of coal gangue and red mud. The soluble clay minerals in the coal gangue and the adsorption alkali in the red mud were eluted in the early stage of leaching. However, in the late stage of leaching, the ANP of the leaching solution was mainly determined by the acidic components produced by the oxidation of pyrite and the bound alkali derived from red mud. The amount of bound alkali leaching depended on the amount of acid produced by the coal gangue oxidation. Therefore, there was no significant difference in the ANP of the leaching solution for the different storage methods.

EC Value

The EC value of the leaching solution varied with the cumulative heating time, as shown in Figure 12.

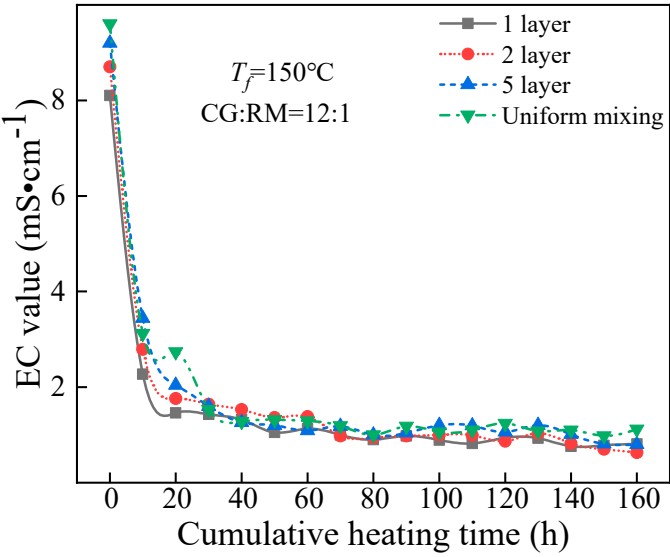

**Figure 12.** EC value variations of the co-disposal leaching solutions corresponding to various storage methods.

Figure 12 shows that the EC value of the leaching solution corresponding to different storage methods had a similar change with the cumulative heating time, that was, it first decreased sharply and then remained stable, and the steady-state value was lower than 1.5 mS/cm. The variation of ANP indicated that the soluble minerals in the co-disposal of coal gangue and red mud under the different storage methods leached rapidly in the early stage.

At $t = 0$ h, the EC value of the initial wetting solution corresponding to the different storage methods reached the maximum, and the maximum value increased as the alternate accumulation layers increased. It also indicated that the increased contact area between coal gangue and red mud

helped to improve the leaching of the clay minerals in the coal gangue and the adsorption alkali in the red mud.

In the late stage of leaching (30–160 h), the different storage methods had no significant effect on the steady-state EC value, for the soluble salt content of coal gangue and red mud had decreased, and the soluble minerals leached were mainly the acidic components obtained from coal gangue and the bound alkali components of red mud. The leaching amount of bound alkali depended on the amount of acid derived from coal gangue. Meanwhile, the acid produced by the oxidation of coal gangue was significantly affected by the sulfur content of coal gangue, the external temperature, and the degree of being contact with the air but had nothing to do with the co-disposal method of coal gangue and red mud.

Considering the operability of the co-disposal of coal gangue and red mud to prevent and control the acid pollution caused by coal gangue, it is proper to adopt the alternative single layer by the coal gangue covered with the red mud during the stacking process.

## 4. Conclusions

(1) The acid production rate of coal gangue is significantly affected by the self-heating temperature. The oxidation degree of pyrite increases as the self-heating temperature increases. However, excessive temperature leads to the reduction in soluble acidic products. In the temperature range of this study, coal gangues had the fastest rate of acid production at 150 °C, the leaching solution presented as strongly acidic and the acid pollution to the environment lasted for a long time.

(2) The addition of red mud can significantly alleviate the acid pollution caused by coal gangue and has long-term effectiveness in the treatment of acid pollution. The mass ratio of coal gangue to red mud has a significant impact on the co-treatment of acid-base pollution. The most appropriate mass ratio was 12:1, and it ensured that the acid pollution index of the leaching solution reached the standard for long-term discharge. However, the co-treatment effect on acid-base pollution was not significantly affected by the storage method. It was more appropriate to adopt the alternate one layer stacking method to simplify the co-disposal of red mud and coal gangue.

(3) The physical and chemical process of covering red mud to prevent acid pollution derived from coal gangue is complex, which involves the gas-solid-liquid three-phase complex chemical environment. Various components are involved in direct and indirect chemical reactions, microbial catalytic oxidation, and displacement reaction. Mutual transformation occurs under the action of chemical reaction, and migration occurs under the action of thermal stress and seepage force field. Thus, the mechanism of co-disposal of coal gangue and red mud used for alleviating the acid-base pollution, and the transformation states of pollutant components, need to be further studied.

**Author Contributions:** Conceptualization, Z.R. and Y.P.; methodology, Z.R. and W.L.; software, Z.R.; validation, Z.R., Y.P. and W.L.; investigation, Z.R., Y.P. and W.L.; data curation, Z.R. and W.L.; writing—original draft preparation, Z.R.; writing—review and editing, Z.R. and W.L.; visualization, Z.R. All authors have read and agreed to the published version of the manuscript.

**Funding:** This research was funded by the Science and Technology Development Program of Yangquan Coal Industry (Group) Co. Ltd., Grant No. YM12066 and the National Natural Science Foundation of China (51974324).

**Conflicts of Interest:** The authors declare no conflict of interest.

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
