# Peer review of "Co-Disposal of Coal Gangue and Red Mud for Prevention of Acid Mine Drainage Generation from Self-Heating Gangue Dumps"

_minerals, doi:10.3390/min10121081_

Round 1
Reviewer 1 Report
///
The manuscript provides information about studying the feasibility coal gangue's disposed with red mud to prevent acid mine drainage pollution. In this paper described 3 types of laboratory experiment:
- Temperature experiment devoted to coal gangue leaching velocities under different temperature conditions. Cumulative heating time was 160 hours.
- Mass ratio test. The coal gangue covered with red mud is stacked in layers with different mass ratios (12:1~3:1). Cumulative heating time was 160 hours.
- Storage Method test. There were used different storage methods (alternating 1, 2 and 5 layers and uniform mixing). Cumulative heating time was 160 hours.
However, I have a major concern. First of all, I didn't see any experiments in-situ in this paper. Is it correct to past "in-situ prevention" in the title of the article?
It is very good theoretical work, but I have many questions about the applicability of experimental data for a real man-made object.
What was the height of the layers in the Mass ratio test?
To what extent can this height be extrapolated for the real volumes of dump rocks?
In the actual conditions of the coal dump, the temperature rises slowly but constantly until spontaneous combustion begins. It is not possible to maintain a constant temperature of 150С in the dump. What is the temperature of the Gangue Hill of Yangquan Coal Industry dump at different depths?
Self-heating of coals is a long-term process; 160 hours is the less than 7 days. What was the total time experiments duration? And the same question about extrapolation results in timeline.
Minor concerns:
Line 89. The title of the Section "Samples Collection" does not match the content. It mainly describes samples preparation.
Line 103. The title of the Section "Material Characterization" does not match the content. It's about chemical-analytical methods.
Line 115. It will be better to change "geophysical" to "physical".
Line 163. Table 2 needs an explanation of abbreviations, only (FCad), (Cad), (Sad) was explained
Line 258. The title of the Section "In-Situ Prevention" does not match the content.
The results and discussion section are mostly a description of the experimental results and lacks in-depth discussion.
Also, it will be good to see chemical reaction equations during the AMD neutralization process
Line 467. Can authors conclude about long-term effect, if experiments were so short?
In general, the authors presented very interesting results that contribute to scientific knowledge and can be useful in practical use.

Reviewer 2 Report
- The biggest shortcoming of this manuscript is its English language. The manuscript needs extensive English Language modifications starting from the Title. “Red Mud Addition’s In-situ Prevention of…” doesn’t make any sense.
- Line 13: “To study the feasibility coal gangue’s disposed with red mud to in-situ prevent AMD pollution” is not grammatically correct and doesn’t make any sense. Should be rewritten.
- Line 29 and 69: “is a kind of” is not a scientifically used phrase.
- Line 62: Always use indirect phrase not a direct one like “we can prevent”
- Line 64: “are to be alleviated by” is not a grammatically correct phrase
- Line 70: It should be “are generated” instead of “will be generated”
- Line 75: It should be “dust generates” instead of “dust is to be generated”
- Line 75: Here you are talking about Red mud then why this sentence about coal gangue has been placed here is not clear “after coal gangue’s air drying and disintegration course”
- Line 93: How particle size can be negative? What do you mean by “-13mm”? It is not scientifically correct.
- Line 94: “Since the pyrite in coal gangue takes the single crystal, disseminated and stripped” is not a grammatically correct phrase
- Line 95: “are used” is not grammatically correct.
- Line 104: What is a “proximate” and “ultimate” analysis? More scientifically accurate words are physical and chemical characterization.
- Line 134: “are subjected to” is not a grammatically correct phrase
- You need to stick to indirect phrases in past tense while describing your methods. Several phrases exist in the manuscript where this common rule has not been followed. Some examples are: “is studied” (Line 136); “are separately packed into” (Line 137); “pump is set to” (Line 140); “process is stopped” (Line 143); “has been collected” (Line 143); “are set at” (Line 144); “is heated” (Line 145); “has been collected” (Line 146); “are detected” (Line 148); “are carried out” (Line 156). These are only few examples. The whole manuscript should be revised to correct these grammatical errors.
- Another biggest shortcoming of this study is absence of replicates of its experiments. Nothing has been mentioned about the number of replicates in the materials and methods section. None of the reported values has any statistical representation either in forms of standard errors or standard deviation. Lack of any statistical analysis make this study very weak and unacceptable for publication.
- Some part of the materials and methods section is written vaguely without providing detailed description. Sentences like “different mass ratios (12:1, 8:1, 5:1 and 3:1) and storage methods (alternating 1 layer, 2 layers, 5 layers and uniform mixing)” do not provide enough information. The ratio mentioned above which part is 12/8/5/3 and which part is 1? The authors left it upon the readers to assume that. Same goes for the alternating layers information.
- “The proximate analysis of the coal gangue samples was conducted according to Chinese standard (GB/T 212-2008)” is not an acceptable explanation for the international readers. Specially when you are working with two unconventional materials (coal gangue and red mud) you need to provide little background information about the methods such as which media was used to dissolve the materials, what mixing ratio was used for pH, and EC analysis etc. There are too many assumptions left for the readers which is not helpful at all.
- The “results and discussions” section is mostly written as a report. Not enough scientific explanations were provided for the observed trends. For example, in Figure 5 why temperature curves behave differently over 150° C should be explained in detail.
- All the tables should have statistical data. How many replicates (n=?), standard deviation/ standard errors should be included against each parameter.
- Same comment goes to all the sections under results and discussion. Without any statistical data none of the outcome can be explained with certainty.
Reviewer 3 Report
This paper describes the neutralization of AMD from coal gangue by the addition of red mud. The authors conducted both batch and column experiments to evaluate the effectiveness of the alkali from red mud. However, several contents are not understandable. I hope the authors revise them dramatically.
- A variety of experimental results are shown. However, there is no description of how to do the experiments and analysis in Materials and methods, such as particle size, shake extraction etc.
- There is no explanation of how to pack the gangue and red mud with uniform layer, single-layer and multi-layers in the reaction column (thickness of each layer, and order of layer). It is difficult to understand the structure of the layers.
- Proximate analysis and ultimate analysis are not understandable.
- Thera are many abbreviations in Table 2. Many of them are not explained.
- The authors use ‘rule’ such as in Line 204. That seems strange. What is rule?
- Underground water the III level is in Fig. 5. This is the regulated pH range? The authors should explain the term.
- There is no explanation of bound alkali and adsorption alkali. The authors should define the difference first.
- The time tense is not understandable. When you conducted the experiments and obtained the data, you should use the past tense. You should clearly distinguish the tense throughout the text.
- Title seems strange. In-situ prevention… by red mud addition?
- The authors should mention on trace elements released from both coal gangue and red mud. I understand that this paper focuses on pH. However, if some trace toxic elements are released from them, we should cover not only pH but also the toxic elements.
- English should be scrutinized again.
The details are followings;
- 13 feasibility -> feasibility of?
- 14 in situ -> Delete.
- 22 ensure -> ensure that
- 23 indexes -> indices
- 84 Why are capital letters used for Automatic …?
- 136 is -> are
- 141 mL.cm-1 seems strange. Check the unit.
- 147 content -> concentration
- 176 50% that it indicates -> 50%. This indicates that
- 183 furthermore -> and?
- 193 mud, -> mud, and
- 196 solution -> solutions
- 198 presents -> present
- 229 mainly -> main
- 261 explores -> explore
- 269 as is -> as
- 272 -273 Check the sentence.
- 395 mainly -> mainly composed?
- 459 The authors should add one line between 4. Conclusion and (1)
- 474 couples -> coupled with?
- 477 migration -> and migration
- 572 Mnamba -> M.?
Round 2
Reviewer 2 Report
The manuscript has been significantly improved by incorporating the reviewer's comments. Still some English Language modifications are needed before the manuscript can be accepted for publication.
- In abstract, Line 23 specify what 12:1 signifies (coal gangue: red mud)
- Line 30: Rewrite the sentence as "Coal gangue is a by-product of coal mining and washing and its discharge amount accounts..."
- Line 31: Remove "as a kind of waste dumps"
- Line 39: Add "and" before "combustion occurring..."
- Line 58: Replace "realized" with "obtained"
- Line 64: Replace "realized" with "obtained"
- Line 78: Remove "the" before "1 t"
- Line 164: Remove "respectively"
- Line 166: Replace "E.g" with "eg."
- Line 182: Replace "E.g" with "eg."
- Line 343: Replace "latter" with "later"
- Line 357: Replace "What's more" with "Additionally"
Reviewer 3 Report
The second manuscript is well revised, and more understandable. The trivial comments are following;
L. 19 and -> and that
L. 30 gangue, -> gangue is; washing, -> washing, and
L. 39 combustion -> and combusion
L. 90 ,which -> , which
L. 117 1mm -> 1 mm I found the same in the other.
L. 123 40kV -> 40 kV; 40mA -> 40 mA
L. 156 showed -> shown
L. 157 600g -> 600 g
L. 163 250mL -> 250 mL I found the same in the other.
L. 166 E. g. -> e. g.
L. 170 test -> tested?
L. 182 methods -> method
L. 211 neutralize with -> neutralize
L. 234 were -> of
L. 238 weak -> low?
L. 252 t should be in italic.
L. 255 meant -> means that
L. 275 present -> are presented
L. 294 gypsum, -> gypsum. ; of which the reason was analyzed to be -> The reason is
L. 295 and part -> whereas part
L. 301 chosen -> was chosen
L. 313 that was -> that is
L. 317 Eq. -> Eqs.
L. 340 for which -> in which
L. 344 was mainly -> due mainly to
L. 357 What's more -> Furthermore
L. 362 that indicated -> . This indicates
L. 366 it indicated that -> Delete.
L. 369 with -> Delete.
L. 387 while -> while they
L. 416 same -> similar
L. 418 process, -> process, and
L. 440 shows -> shows that
L. 477 mainly were -> were mainly
L. 497 is -> was
L. 498 is -> was
L. 500 process -> process of
